**Data Availability Statement:** The sequencing data has been deposited at the Sequence Read Archive (SRA) under the BioProject ID PRJNA635454. All

# Signatures of somatic mutations and gene expression from p16INK4A positive head and neck squamous cell carcinomas (HNSCC)

**Nabil F. Saba**[1☯]*, **Ashok R. Dinasarapu**[2☯], **Kelly R. Magliocca**[1], **Bhakti Dwivedi**[1], **Sandra Seby**[3], **Zhaohui S. Qin**[4], **Mihir Patel**[1], **Christopher C. Griffith**[5], **Xu Wang**[1], **Mark El-Deiry**[1], **Conor Ernst Steuer**[1], **Jeanne Kowalski**[6], **Dong Moon Shin**[1], **Michael E. Zwick**[2], **Zhuo Georgia Chen**[1,7]*

1 Winship Cancer Institute, Emory University, Atlanta, GA, United States of America, 2 Department of Human Genetics, Emory University, Atlanta, GA, United States of America, 3 Centers for Disease Control (CDC), Atlanta, GA, United States of America, 4 Department of Biostatistics and Bioinformatics, Emory University Atlanta, Atlanta, GA, United States of America, 5 Department of Pathology, Cleveland Clinics, Cleveland, OH, United States of America, 6 Department of Oncology, Dell Medical School, Austin, Texas, United States of America, 7 Department of Hematology and Medical Oncology, Emory University Atlanta, Atlanta, GA, United States of America

☯ These authors contributed equally to this work.
* gzchen@emory.edu (ZGC); nfsaba@emory.edu (NFS)

## Abstract

Human papilloma virus (HPV) causes a subset of head and neck squamous cell carcinomas (HNSCC) of the oropharynx. We combined targeted DNA- and genome-wide RNA-sequencing to identify genetic variants and gene expression signatures respectively from patients with HNSCC including oropharyngeal squamous cell carcinomas (OPSCC). DNA and RNA were purified from 35- formalin fixed and paraffin embedded (FFPE) HNSCC tumor samples. Immuno-histochemical evaluation of tumors was performed to determine the expression levels of p16INK4A and classified tumor samples either p16+ or p16-. Using ClearSeq Comprehensive Cancer panel, we examined the distribution of somatic mutations. Somatic single-nucleotide variants (SNV) were called using GATK-Mutect2 ("tumor-only" mode) approach. Using RNA-seq, we identified a catalog of 1,044 and 8 genes as significantly expressed between p16+ and p16-, respectively at FDR 0.05 (5%) and 0.1 (10%). The clinicopathological characteristics of the patients including anatomical site, smoking and survival were analyzed when comparing p16+ and p16- tumors. The majority of tumors (65%) were p16+. Population sequence variant databases, including gnomAD, ExAC, COSMIC and dbSNP, were used to identify the mutational landscape of somatic sequence variants within sequenced genes. Hierarchical clustering of The Cancer Genome Atlas (TCGA) samples based on HPV-status was observed using differentially expressed genes. Using RNA-seq in parallel with targeted DNA-seq, we identified mutational and gene expression signatures characteristic of p16+ and p16- HNSCC. Our gene signatures are consistent with previously published data including TCGA and support the need to further explore the biologic relevance of these alterations in HNSCC.

other relevant data are within the paper and its Supporting Information files.

**Funding:** This research was supported by a grant NCI R21 CA182661-01A1 to NFS (5R21CA182661-02) and GZC (GZC 5R21CA182661-02). The funders had no role in study design, data collection and analysis, decision to publish, or preparation of the manuscript.

**Competing interests:** No Authors have competing interests.

**Abbreviations:** HNSCC, Head and Neck squamous cell carcinomas; COSMIC, Catalog of Somatic Mutations in Cancer; dbSNP, database of single nucleotide polymorphisms; HAPMAP, The haplotype map database; FFPE, stands for formalin-fixed, parafin-embedded; ExAC, Exome Aggregation Consortium; gnomAD, The Genome Aggregation Database; GATK, Genome Analysis Toolkit; TCGA, The Cancer Genome Atlas project; OPSCC, Oropharyngeal Squamous Cell Carcinoma; p16+, p16$^{INK4A}$ overexpression; HPV, Human Papilloma Virus.

## Introduction

Recent reports have noted an increase in the incidence of head and neck squamous cell carcinomas (HNSCC), which account for approximately 3% of all cancers in the US (https://seer.cancer.gov/). These cancers are more than twice as common among men as among women. Tobacco and alcohol use are the major risk factors for HNSCC (~75%), while infection with high-risk types of human papilloma virus (HPV) accounts for the remainder of HNSCC [1–3]. HNSCC, particularly oropharyngeal squamous cell carcinomas (OPSCC), that are caused by infection with cancer causing types of HPV, especially HPV type 16, show distinct clinical, pathological, and molecular features compared with non-HPV related cancer [1,4]. OPSCC and non-OPSCC are two subsites with in HNSCC. Increased expression of p16$^{INK4A}$, referred to hereafter as p16+, has been reported to strongly correlate with HPV infection in HNSCC; however, p16-positivity is not limited to HPV-positive tumors and therefore, is not a perfect surrogate for HPV positivity [5,6]. Previous studies have indicated that in HNSCC, HPV-positive patients have better overall survival cure rates than their HPV-negative counterparts [7].

Multiple studies have identified significant enrichment of somatic mutations in genes such as *PIK3CA*, *TP53*, *CDKN2A*, *FGFR3*, *PTEN and RB1* in HPV-associated HNSCC [4,8–10]. Other distinctions between HPV-positive and HPV-negative cancers have been identified by combined analysis of somatic mutations and gene expression profiles [11,12]. Although the ClearSeq Comprehensive Cancer panel can identify driver mutations in the coding region of 151 cancer genes, it may fail to detect structural changes, such as gene fusions, that may be therapeutically relevant. The possibility of therapeutically actionable gene fusions driving HNSCC has not been fully explored. We therefore proposed that a combination of RNA-Seq and targeted DNA-seq to identify a high yield diagnostic tailored for HNSCC. In addition, we aimed to correlate our methods with those used in other published studies in HNSCC. A prior proof-of-principle study using a small sample size confirmed the feasibility of using FFPE samples in HNSCC to that end [13]. Here, we report an analysis of genetic alterations in HNSCC arising in diverse anatomical sites by use of targeted DNA-seq and RNA-seq of FFPE tumor samples mainly from patients with OPSCC.

## Results

### Somatic mutation analysis and confirmation of recurrent gene mutations among HNSCC individuals

We conducted targeted sequencing of 151 disease-associated genes that have been implicated in studies of a wide range of cancers in 22 p16+ (HPV+), 4 p16- (HPV-), and one p16-unknown (HPV unknown status) tumor samples using Agilent's ClearSeq Comprehensive Cancer panel. Clinical and demographic information are provided in **Table 1 and S1 Table**. A summary of somatic mutations in p16+ and p16- tumor samples is reported in **Figs 1 and 2, and S2 Table**. We restricted our analysis to predicted protein coding gene regions and excluded synonymous and noncoding variants from our analysis (**S3 Table**). We observed C>T substitutions were increased in p16+ as compared to p16- cancers (**Fig 2**; [4]). The most common mutations were in *PIK3CA*, *KMT2A*, and *PTEN*. We also identified that *PIK3CA* mutations were localized to E542K, E545K, and H1047L hotspots known to promote activation, with the remaining mutations of uncertain function (**Fig 1C and 1D**; [14]). Many canonical pathways known to be involved in oncogenic signaling were mutated, including RTK-RAS and PI3K ([15]; **Fig 3**). Further, the "druggability" of the 25 genes specifically mutated in the RTK-RAS and PI3K pathways was searched using the Drug Gene Interaction database (DGIdb) [16] and the majority of the genes were found to be in druggable categories (**S1 Fig**).

**Table 1. Demographic and clinical characteristics of study population.**

|  | p16+ | p16- | p16 unknown |
|---|---|---|---|
| **N (% sample)** | 23 (66) | 5 (14) | 7 (20) |
| **Gender** | | | |
| Male | 22 | 2 | 4 |
| Female | 1 | 3 | 3 |
| **Age** | | | |
| Range | 41–72 | 50–71 | 29–74 |
| Mean | 58.3 | 62.4 | 58.9 |
| **Tobacco** | | | |
| Current | 3 | 2 | 1 |
| Former | 11 | 2 | 5 |
| Never | 8 | 1 | 1 |
| Other (Cannabis) | 1 | 0 | 0 |
| **Therapy** | | | |
| Radiation (Yes; No; Unknown) | 18; 4; 1 | 1; 4; 0 | 4; 3; 0 |
| Chemo (Yes; No; Unknown) | 18; 4; 1 | 3; 2; 0 | 3; 4; 0 |
| **Anatomy** | | | |
| Tongue | 0 | 1 | 5 |
| FOM | 0 | 0 | 1 |
| BOT | 11 | 3 | 0 |
| Tonsil | 10 | 1 | 0 |
| Larynx | 0 | 0 | 1 |
| Unknown | 2 | 0 | 0 |

BOT–Base of Tongue; FOM–Floor of Mouth; p16+—Over expression of p16$^{INK4A}$.

Among these were 5 genes for which "clinically actionable" compounds are available including the EGFR and PIK3CA related genes.

## Gene fusion detection

In HNSCC, TCGA provided the first report on gene fusions [14]. Among these fusion events, a known gene fusion, FGFR3-TACC3, was identified in two p16+ tumors i.e GHN-66 and GHN-80 (**S4 Table**). Other studies also showed that gene fusions are associated with significant upregulation of genes including EGFR [18]. However, we did not see the upregulation of EGFR gene expression.

## Differential expression of genes among HNSCC individuals

We conducted differential expression analysis between p16+ (N = 12) and p16- (N = 5), or p16+ (N = 12) and p16-unknown (N = 7) groups and created a catalog of gene expression alterations (**Fig 4A–4C**). The number of differentially expressed (DE) genes in p16+ vs p16- comparison was 1,044 and 8, respectively at FDR < 0.05 and 0.01 (total of 16,178 genes assayed). Among a total of 16,253 tested genes, the number of differentially expressed (DE) genes in p16+ vs p16-unknown group comparison was 574 and 58, respectively at FDR < 0.05 and 0.01. To confirm the findings that we obtained using our dataset, we used the TCGA dataset for validation. Specifically, we used comparable tissues of origin including the base of tongue (BOT), tonsil etc (**S5 Table**). The DEGs identified in our analyses were used to create a heatmap with TCGA expression data (**Fig 4D**). With this abundance of gene expression data, we have

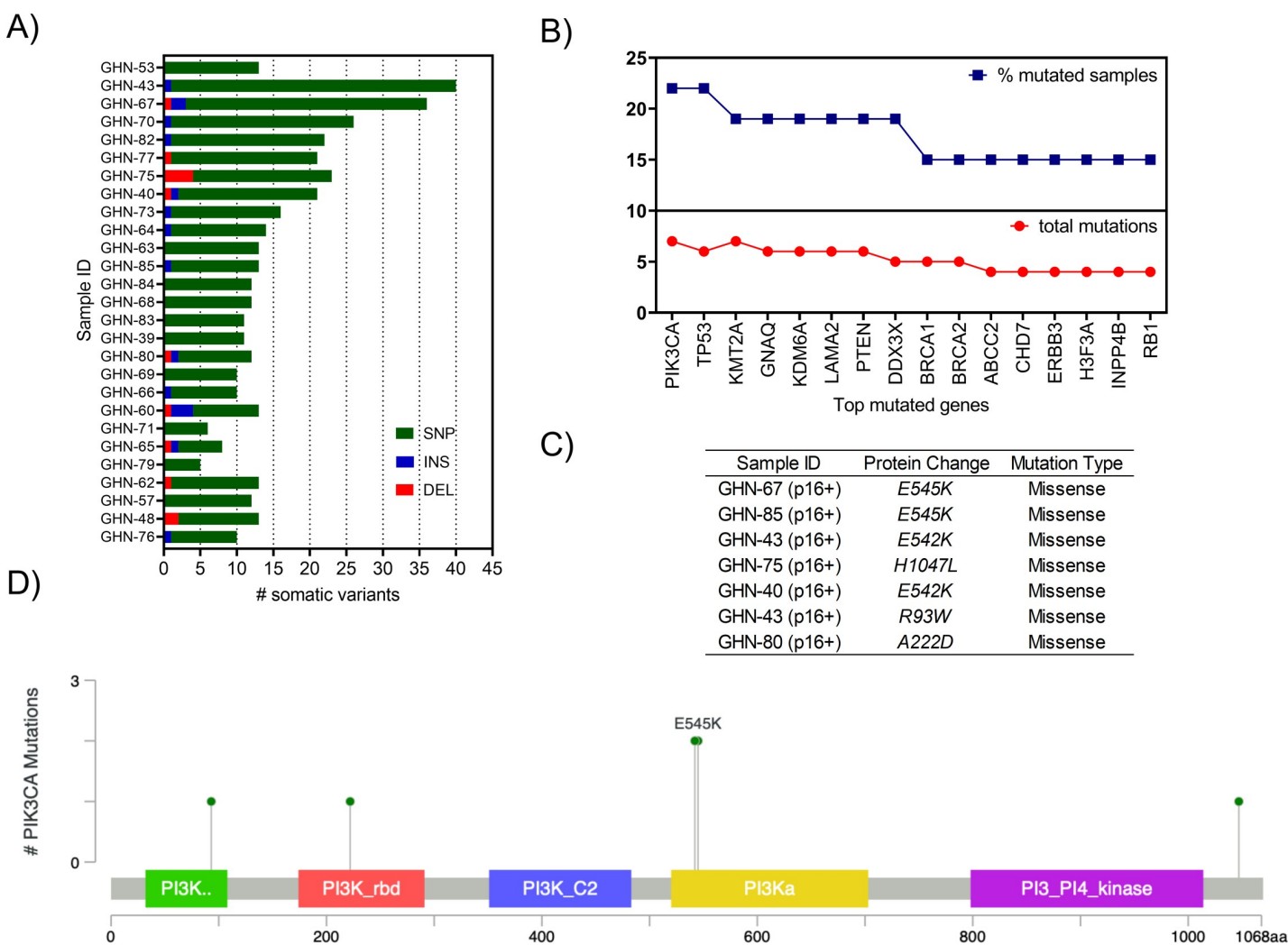

**Fig 1. Summary of somatic mutations identified.** A) Barplots showing numbers of mutations detected in the HNSCC cohort using GATK tumor-only method after each filtering step (see Methods section). MAFtools was used to generate the list of top mutated genes from 27 subjects, including p16+, p16- and p16-unknown status. B) Top 15 mutated genes. C) PIK3CA mutations identified in p16+ samples. Schematic of protein domains, displaying sites of mutations identified in the most frequently mutated gene PIK3CA in p16+ tumor samples.

identified a misclassified TCGA sample. The sample TCGA-CV-5971 was identified by TCGA as HPV+ but our gene expression data show that its expression is similar to that of the HPV-group (**Fig 4D** & **S6 Table**). Canonical pathway analysis on the differentially expressed genes (p16+ vs p16-) using ingenuity pathway analysis (IPA) revealed most significant enrichment in pathways related to cell cycle (**Fig 5**).

### p16 expression and survival

Survival analyses were performed using the Kaplan-Meier method with the log-rank test for statistical significance. From the curves, it is evident that the patients, who have negative (or unknown) status for the p16, have more death rate as compared to the patients, who are positive for p16. For all 3 groups, the rate of decrease in survival rate is fairly constant (**Fig 6**). Moreover, a clear association of Kaplan-Meier survival curve of p16-based molecular subtype p16-negative with p16-unknown group supports clustering of RNA-seq based gene expression

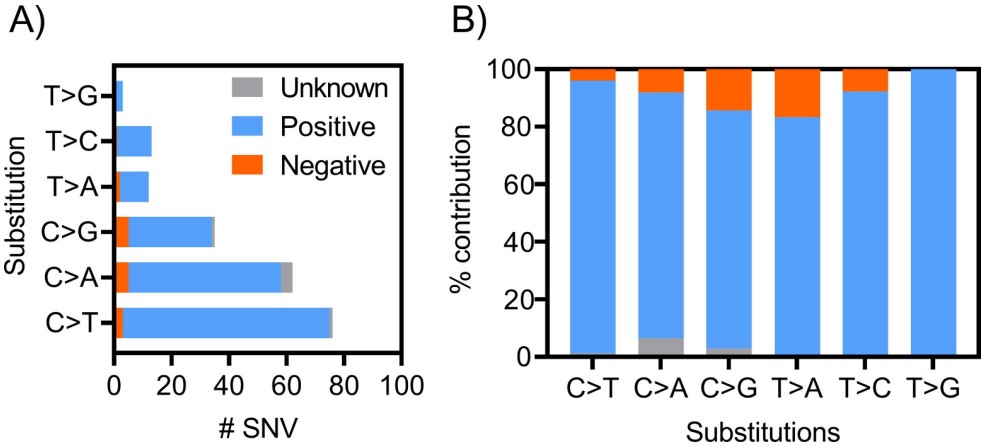

**Fig 2. Summary of SNV identified. A**) Individual number of SNVs identified from all 3 classes of HNSCC samples. For example, 72, 3 and 1 C>T substitutions were identified from 22 p16+, 4 p16- and 1 p16-unknown samples, respectively **B**) Percent contribution of individual SNVs from all 3 classes of HNSCC samples.

profiles (**S2 Fig**). As the p16 unknown group consisted of non-oropharyngeal tumors, these results were not surprising as the significance of p16 status in non-oropharyngeal primaries is unclear.

## Discussion

Using RNA-seq data in parallel with targeted sequencing of a panel of 151 genes, we demonstrate that gene expression data from FFPE samples can identify gene signatures characteristic of p16 + versus p16- OPSCC which is a subsite of HNSCC. These signatures need to be further explored for their biologic relevance in OPSCC. One caveat is the relatively limited size of the p16-negative group, which poses challenges in achieving accurate estimates of biological variability and statistical robustness in data analysis. Although matched tumor-normal analysis is preferred due to higher precision, we demonstrate that mutation detection without matched normal samples is possible for certain applications. We were also able to confirm the reliability of using routine FFPE specimens to accurately identify possible targeted pathways that were confirmed relevant in p16-positive versus p16-negative tumors in prior reports, and that could have wider future applicability and inform clinical applications in HNSCC.

## Materials and methods

### Research subjects

The study was approved by Emory University Institutional Review Board (IRB). Study participants included individuals with a pathologically documented diagnosis of HNSCC/OPSCC and who had undergone surgical resection of their primary disease or had adequate core biopsies of their primary tumors. A summary of clinical data of the study participants is shown in **Table 1** and **S1 Table**. There were 35 samples total; 27 were used for DNA-Seq and 26 for RNA-Seq with 16 samples overlapping. All tissue samples were collected from subjects who gave written and signed informed consent using the Head and Neck Cancer Winship-Emory University IRB-approved consent form for tissue collection. Tissue samples were collected and stored at the Emory University Winship Cancer Institute, in Atlanta, Georgia. Clinical data collected included gender, smoking history, radiation treatment status, chemotherapy treatment status and stage (AJCC 7). A total of 35 formalin-fixed paraffin embedded (FFPE)

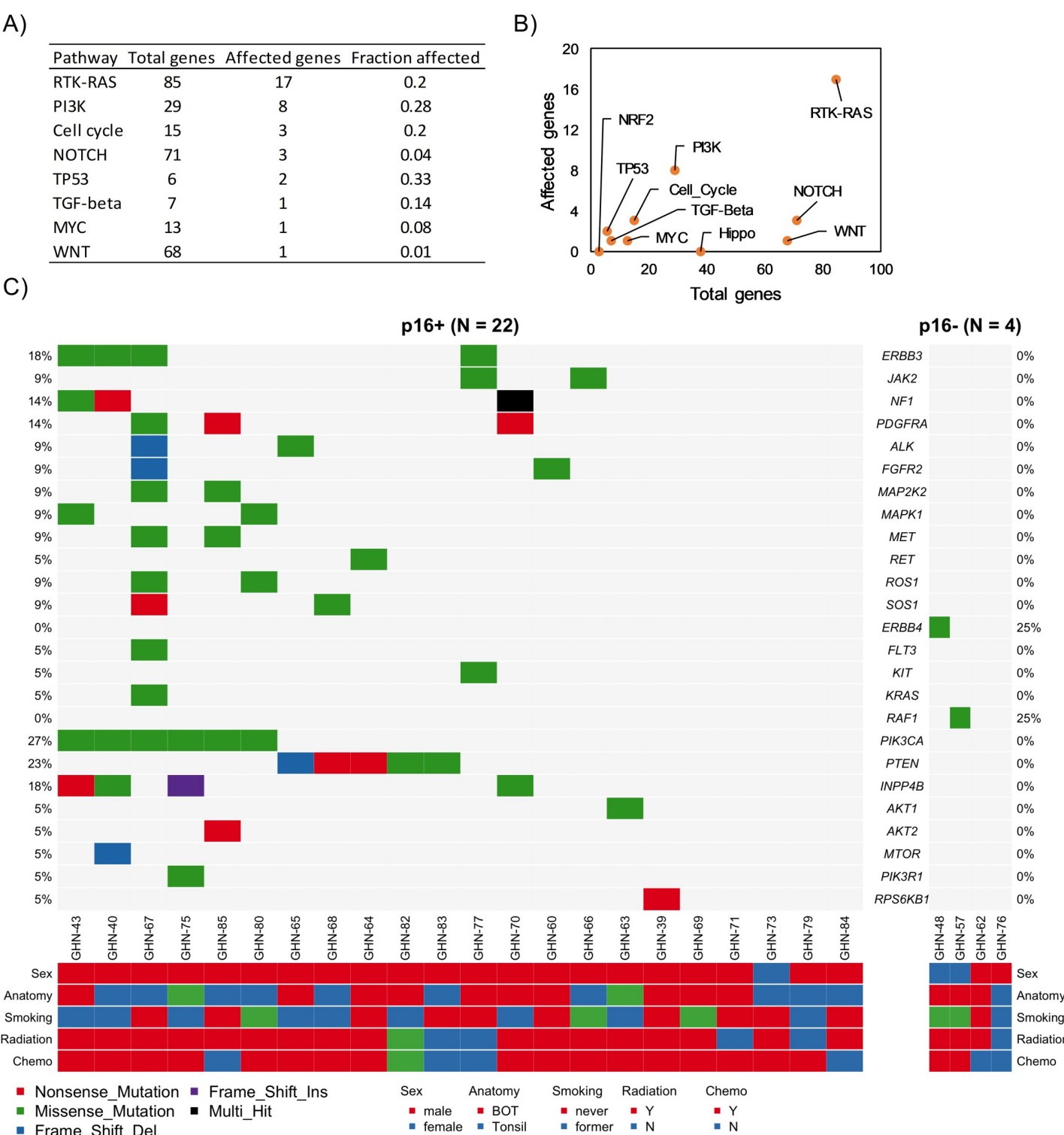

**Fig 3. Enrichment of known oncogenic pathways based on 26 HNSCC samples including p16+ (N = 22) and p16- (N = 4) groups.** Oncogenic signaling pathways are derived from TCGA cohorts [17]. **A-B**) Mutation affected pathways **C**) 25 mutated genes of the RTK-RAS and PI3K signaling pathways (22 p16+ and 4 p16-). See **S1 Fig** for their known/reported drug-gene interactions and druggable categories. All gene-drug interactions and drug claims are compiled from the Drug Gene Interaction Database [16].

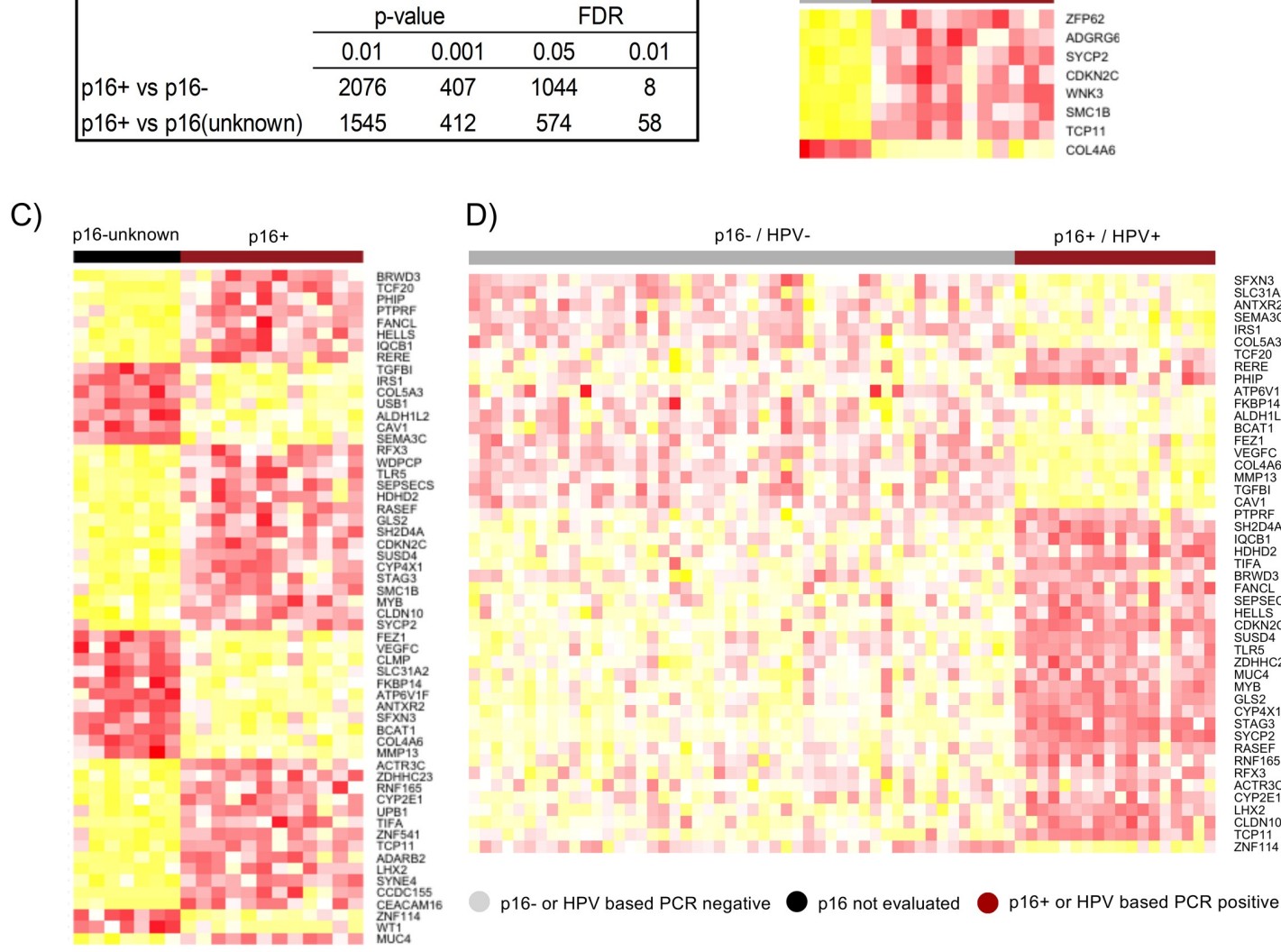

**Fig 4. Differentially Expressed Genes (DEGs) from a cohort of HNSCC. A**) Number of DEGs in p16+ vs p16- and p16+ vs p16-unknown comparisons **B-C**) Heatmaps of 8 and 58 DEGs with FDR<0.01, respectively, between p16+ vs p16- and p16+ vs p16-unknown **D**) Heatmap generated based on TCGA [14] HNSCC data confirming the gene expression consistency. Yellow indicates reduced expression while red indicates increased expression.

OPSCC tumor specimens, 10 of which were oral cavity specimens were included in the study. As part of the routine pathology diagnosis of OPSCC, the tumors were evaluated for p16 immunoreactivity. Some non-oropharyngeal tumors had exhausted tissue and could not be checked for p16 (p16 unknown). The FFPE tumor blocks were de-identified according to the IRB-approved protocol. 5μM serial sections of FFPE samples were obtained from each block for DNA and RNA isolation.

## p16 immunohistochemical (IHC) staining

HPV infection status of OPSCC samples was identified using p16 expression in the tumor cells. In brief, p16 ink4a immunohistochemical staining was performed in a standard manner per supplier's instructions (CINtec 9517, MTM Laboratories, Westborough, MA) on FFPE

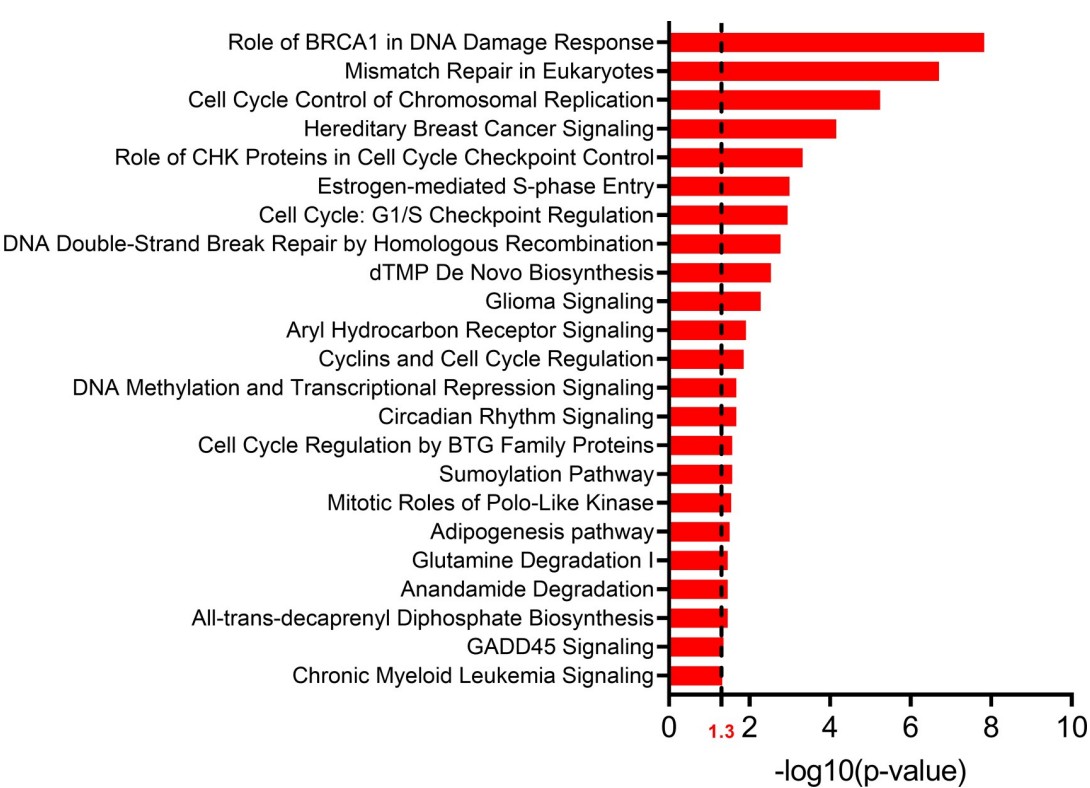

**Fig 5. Significantly affected pathways (p $<$ = 0.05) based on Ingenuity Pathway Analysis (IPA).**

tissue sections using the automated, open system immunostainer (DAKO AutoStainer Link 48, Copenhagen, Denmark). The slides were processed using the DAB reagent to visualize the antibody-antigen complex, then counterstained with hematoxylin and subsequently washed and cover slipped. Both positive and negative control slides were prepared. The proportion of tumor cells demonstrating nuclear and cytoplasmic p16 staining were categorized dichotomously as either p16INK4A-positive ($>$ 70% tumor cells exhibiting strong and diffuse nuclear and cytoplasmic staining) p16INK4A-negative ($<$70% tumor cells exhibiting strong and diffuse nuclear and cytoplasmic The IHC staining for expressed p16 was performed on five-micron sections of FFPE tissue sections. A pathologist (K.M) verified clinical diagnosis and HPV status of each specimen.

## DNA and RNA extraction

Genomic DNA and total RNA were isolated using Omega BioTek chemistries according to the manufacturer's protocol from 5 μM sections for each isolate. DNA was quantitated using NanoDrop and Qubit, and RNA was quantitated using NanoDrop and Agilent BioAnalyzer.

## RNA-seq library preparation

The RNA from FFPE tumor samples was used to prepare libraries for RNA sequencing. 200 ng of genomic DNA from each specimen was used to generate uniquely barcoded sequencing libraries. Briefly, libraries were prepared using the Ion DNA Barcoding and Ion Xpress Template kits (Life Technologies, Grand Island, NY, USA) according to the manufacturer's protocols. Fragment sizes were assessed on an Agilent Bioanalyzer 2100 using the High Sensitivity

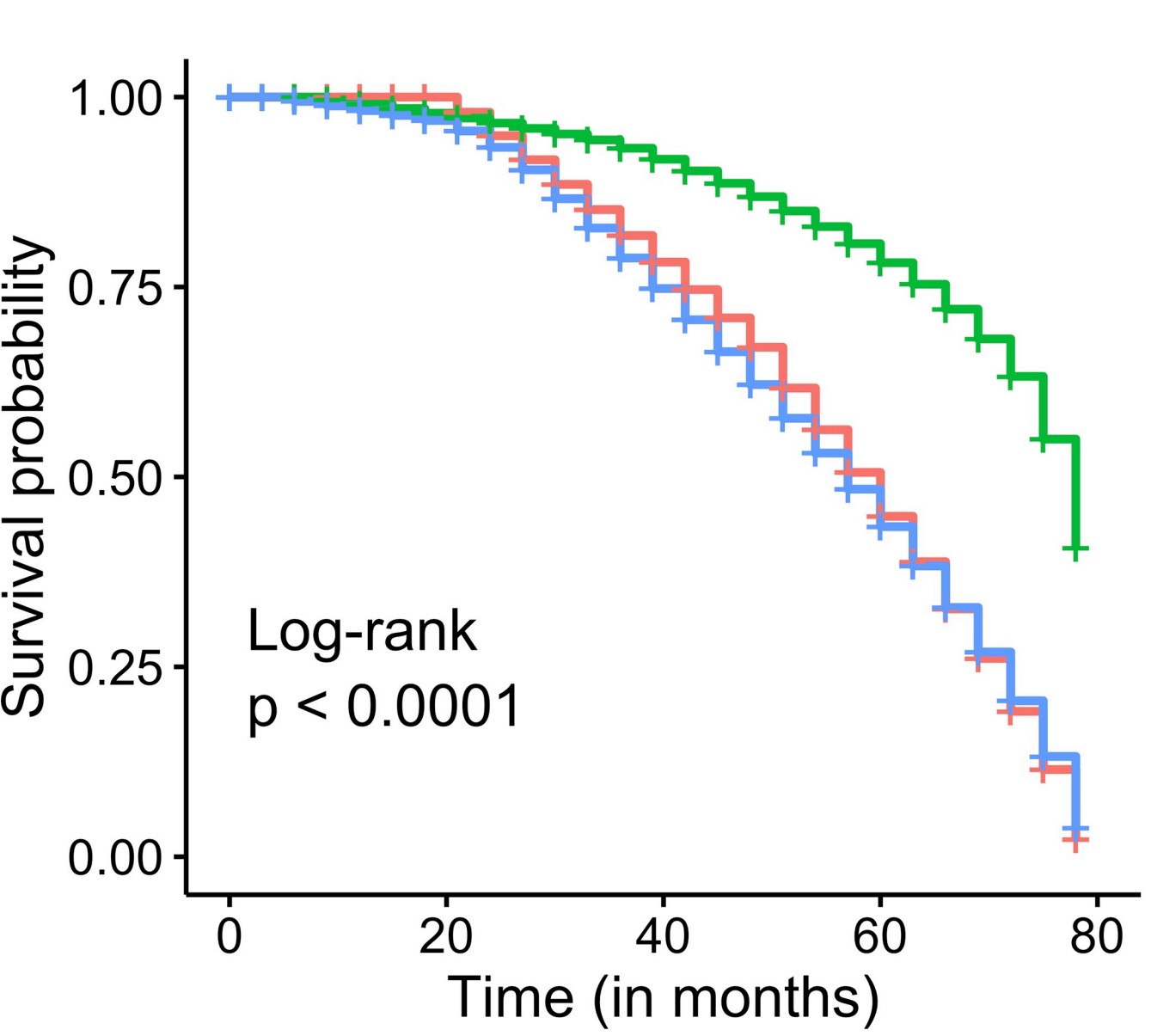

**Fig 6. Kaplan-Meier curves of the molecular subtypes of HNSCC cohort based on p16 status.**

kit (Agilent, Santa Clara, CA, USA). All 8 uniquely barcoded specimen libraries were run on a single 318 chip and sequenced using an Ion PGM 200 sequencing kit.

### Next-generation comprehensive cancer profiling

Agilent's targeted sequencing ClearSeq Comprehensive Cancer Panel was used. We extracted DNA from selected samples (N = 27) and sequenced each sample for 151 disease-associated genes that have been implicated in studies of a wide range of cancers. All coding exons, exon-intron boundaries and selected introns of these genes are targeted. Libraries were paired-end

sequenced using the Illumina MiSeq platform with the 75bp paired-end read mode. On aver-age 10 million reads per sample were generated.

## Short-read mapping, and somatic variant calling

Reads were trimmed for adaptors and paired-end mapped to the reference human genome (hg19) using BWA-MEM algorithm (version 0.7.12) from GenAligners v3.0 (SureCall 4.0, analysis software from Agilent Technologies). For the detection of somatic single-nucleotide polymorphism (SNP) and insertion and deletion (indel), we used Mutect2 (GATK v4) on tumor samples ("tumor-only" mode) on each sample. Resulting BAM files were sorted using Samtools v1.3 and PCR duplicates were marked using Picard v2.6.0. Realignment was per-formed following the Genome Analysis Toolkit (GATK) best practices. A base quality recali-bration table was generated using GATK BaseRecalibrator with one or more databases of known polymorphic sites (dbSNP138 (hg19) and HAPMAP 3.3 (hg19) from the GATK resource bundle). The appropriate liftover chain file from GATK resource bundle also down-loaded (b37tohg19.chain) to convert the genomic coordinates from b17 to hg19 build. To restrict a subset of genomic regions in variant calling while using GATK tools, we have pro-vided a.bed file of exome intervals obtained from Agilent website. VCF containing population allele frequencies (AF > 0.05) of common germline variants from ExAC and an exome inter-vals (.bed) file were provided to GATK GetPileupSummaries to get a summary of read counts from recalibrated tumor BAM that support a set number of known variant sites. This summary of read counts was used to calculate cross-sample contamination. To call somatic variants from tumor (recalibrated) BAM via local assembly of haplotypes (using Mutect2), a VCF con-taining population allele frequencies of common and rare alleles from gnomAD to avoid call-ing any germline variants and a.bed file of exome intervals were used. The Mutect2 called variants were then filtered based on the contamination estimate using GATK *FilterMutectCalls* and only passed somatic variants were included in the further analysis. Calls that are likely true positives get the PASS label in the FILTER field. This step seemingly applies 14 filters, includ-ing contamination.

## Comparison of mutations with COSMIC database

To determine if the called variants have been previously detected, we have annotated the derived list of somatic variants using Catalog of Somatic Mutations in Cancer (COSMIC) ver-sion 87 [19] (hg19, Coding and Non Coding vcf files combined) with ANNOVAR tool [20]. Mutations not detected by the above methods were considered novel (**S3 Table**).

## Analysis of significantly mutated pathways

We examined the distribution of mutations in known oncogenic signaling pathways derived from TCGA cohorts] 17]. The mutation profiles in p16+ and p16- were shown with the R package "MAFtools" [21]. We also used MAFtools to calculate the mutation rate of each gene.

## Data pre-processing and alignment of sequenced reads

Quality assessment of raw FASTQ reads was performed using the FASTQC program. Paired end RNA-Seq samples were mapped to the human genome reference assembly (hg38) with STAR 2.4.2a. Transcript expressions as counts were estimated with HTSeq. Count data were subsequently normalized using TMM (weighted trimmed mean of M-values) with the EdgeR package [22], and converted to counts per million (CPM) and log2-transformed. A filtering process was also performed to exclude the genes without at least 10 counts in 33% of the

samples. For gene-fusion detection, we use STAR-Fusion (https://github.com/STAR-Fusion/STAR-Fusion66). It is a method that accurately identifies fusion transcripts from RNA-seq data and outputs all supporting data discovered during alignment.

## Data visualization and clustering analysis

The similarity of the relative gene transcript abundances (using log2-transformed values of CPM) for each of the samples was compared using an unsupervised hierarchical clustering and heatmap analysis in R. Unsupervised hierarchical clustering of the differentially expressed genes was performed using Pearson correlation distance and average clustering.

## Differential gene expression analysis

To assess the significance in the difference between p16-positive OPSCC samples and p16-negative OPSCC samples in terms of gene expression, we used the two-sample t-test. Transcripts were considered to be differentially expressed if their FDR < 0.05. Volcano plots of -log10(p-value) vs. log2 (CPM) fold-change were made to examine these associations in each tissue pair within each individual.

## Analysis of significantly enriched pathways

WEB-based Ingenuity Pathway Analysis (IPA) (QIAGEN) was used for pathway analysis to identify pathways that were enriched in all significant gene lists by each of the HPV pairs (p16-positive vs. p16-negative). Only genes that were differentially expressed in sample comparisons (significance at p-value ≤ 0.05) were included in the analysis. Pathways were considered to be significant if the pathway's p-value of enrichment was ≤0.01.

## The Cancer Genome Atlas (TCGA)

TCGA RNA-seq data including 49 HPV-negative and 18 HPV-positive tissue samples in the form of raw gene count (disease ="HNSC" and data.type ="RNASeq2") was downloaded using TCGA2STAT package for R [23] and used to find overlap between TCGA gene expression and our HNSCC data. Both expression data and clinical data were available for 67 HNSCC tumors (from different locations: oral tongue, BOT and tonsil). We used the dataset abbreviations as defined by the TCGA consortium, a public resource that catalogues clinical data and molecular characterizations of many cancer types (as defined in https://tcga-data.nci.nih.gov/docs/publications/tcga/).

In order to validate the differentially expressed gene signature identified using our OPSCC patient samples, we queried transcriptome data for HNSCC downloaded from the TCGA cancer program. Only the patient tumor samples with available clinical and RNA-seq gene expression data were obtained. As our study focuses only on oropharyngeal samples, patient samples obtained from anatomic sites—"Base of Tongue" and "Tonsil" with p16 status annotation were only used in this analysis, resulting in a total of 67 samples (p16-positive OPSCC samples (n = 18) and p16-negative OPSCC samples (n = 49)). The remaining samples were excluded as they were from a different anatomic site or p16 status annotation was not available. Unsupervised hierarchical clustering of the OPSCC TCGA patient samples was performed based on the expression of our differentially expressed gene signature using Pearson correlation distance and average clustering.

## Gene fusion analysis

A fusion gene refers to two genes (either in whole or in part) that undergo fusion resulting in a chimeric gene, which is usually caused by reasons such as chromosome translocation and associated problems. STAR-fusion software analysis and the detection of fusion genes were used to identify fusion genes. Fusion gene lists were filtered with STAR-fusion with the default filter method and other parameters.

## Statistical analysis

Statistical analyses were conducted using R (version 4.0.1; https://www.r-project.org/). Survival analysis was performed via "survival" and "survminer" R packages [https://github.com/therneau/survival and https://github.com/kassambara/survminer/]. The overall survival (OS) was evaluated by Kaplan-Meier curves and the statistical difference was estimated using log-rank test.

## Supporting information

**S1 Table. Tumor DNA-seq (ClearSeq, N = 27) and RNA-seq (N = 24) samples.**
(DOCX)

**S2 Table. Summary of somatic variant annotations from Annovar.**
(DOCX)

**S3 Table. Summary of somatic variant annotations from COSMIC database.**
(DOCX)

**S4 Table. Fusion genes found in HNSCC samples.**
(XLSX)

**S5 Table. RNA-seq dataset downloaded from TCGA database.**
(DOCX)

**S6 Table. Samples were classified as HPV-positive using an empiric definition of $> 1,000$ mapped RNA-seq reads.**
(DOCX)

**S1 Fig. Following plot shows potential druggable gene categories along with up to top 5 genes involved in them.**
(TIF)

**S2 Fig. Heatmap of DEGs identified between p16+ vs p16- and p16+ vs p16-unknown groups.**
(TIF)

## Acknowledgments

We acknowledge the contribution of Dr Michael Rossi to this work.

## Author Contributions

**Conceptualization:** Nabil F. Saba, Zhuo Georgia Chen.

**Data curation:** Nabil F. Saba, Xu Wang, Mark El-Deiry, Jeanne Kowalski.

**Formal analysis:** Nabil F. Saba, Ashok R. Dinasarapu, Bhakti Dwivedi, Zhaohui S. Qin, Mihir Patel, Conor Ernst Steuer, Jeanne Kowalski, Dong Moon Shin, Michael E. Zwick.

**Funding acquisition:** Nabil F. Saba, Zhuo Georgia Chen.

**Investigation:** Nabil F. Saba, Jeanne Kowalski, Michael E. Zwick, Zhuo Georgia Chen.

**Methodology:** Nabil F. Saba, Ashok R. Dinasarapu, Kelly R. Magliocca, Sandra Seby, Christopher C. Griffith, Michael E. Zwick, Zhuo Georgia Chen.

**Project administration:** Nabil F. Saba, Conor Ernst Steuer.

**Resources:** Nabil F. Saba, Ashok R. Dinasarapu, Mark El-Deiry, Zhuo Georgia Chen.

**Software:** Ashok R. Dinasarapu.

**Supervision:** Nabil F. Saba, Zhuo Georgia Chen.

**Validation:** Ashok R. Dinasarapu.

**Writing – Review & Editing:** Ashok R. Dinasarapu.

**Writing – original draft:** Nabil F. Saba, Ashok R. Dinasarapu.

**Writing – review & editing:** Nabil F. Saba.

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
