## [Decision Letter · Decision Letter 0]

21 May 2020

PONE-D-20-10368

Signatures of somatic mutations and gene expression from p16INK4A positive head and neck squamous cell carcinomas (HNSCC)

PLOS ONE

Dear Dr. Saba,

Thank you for submitting your manuscript to PLOS ONE. After careful consideration, we feel that it has merit but does not fully meet PLOS ONE’s publication criteria as it currently stands. Therefore, we invite you to submit a revised version of the manuscript that addresses the points raised during the review process.

We look forward to receiving your revised manuscript.

Kind regards,

Hyunseok Kang, MD, MPH

Academic Editor

PLOS ONE

Additional Editor Comments (if provided):

Thank you for submitting this work to PLOS ONE. Please consider revising the manuscript as suggested by the reviewers.

2. Please provide additional details regarding participant consent. In the ethics statement in the Methods and online submission information, please ensure that you have specified what type of consent you obtained from patients for tissue collection (for instance, written or verbal, and if verbal, how it was documented and witnessed).

In addition, please provide the source of the tissue samples used in this work (e.g. hospital, institution or medical center name).

3. Please provide the accession number of specific URL weblink of the specific RNA-Seq dataset downloaded from the TCGA for this study.

6. Please upload a new copy of Figure 3 and 6 as the detail is not clear. Please follow the link for more information: https://blogs.plos.org/plos/2019/06/looking-good-tips-for-creating-your-plos-figures-graphics/

Reviewers' comments:

Reviewer's Responses to Questions

**Comments to the Author**

1. Is the manuscript technically sound, and do the data support the conclusions?

Reviewer #1: Yes

Reviewer #2: Partly

2. Has the statistical analysis been performed appropriately and rigorously? 

Reviewer #1: Yes

Reviewer #2: No

3. Have the authors made all data underlying the findings in their manuscript fully available?

Reviewer #1: Yes

Reviewer #2: No

4. Is the manuscript presented in an intelligible fashion and written in standard English?

Reviewer #1: Yes

Reviewer #2: Yes

5. Review Comments to the Author

Reviewer #1: This work presents the signatures of somatic mutation and gene expression from p16ink4a positive HNSCC. Thank you for your valuable research on genetic variants and gene expression signatures of significant HPV infection in head and neck cancer. However, while reviewing this manuscript, it was difficult to check all the details because all the resolutions of the figures were low. Attaching high resolution figures will be helpful in the next review.

In this study, p16 of head and neck cancer FFPE was classified by IHC staining. In case of ‘p16 unknown’, didn't you perform IHC? Or did you stain IHC and the staining result wasn't clear? I recommend you need to define p16? group. I am still not sure why you compared p16 separately in this study. Therefore, the reason for comparing the p16 unknown group with the p16 positive group in this study should be clearly described in the result section.

In addition, in the various experiments conducted in this study, some were targeted only to OPSCC, and some were confirmed in HNSCC, including cancers of a few other origins. Personally, I think it might have been clearer for this study to focus only on OPSCC. Or, in some experiments, please add the reason why only the OPSCC group was tested separately.

Reviewer #2: comments:

1. figures are fuzzy, hardly to read

2. Please explain what you found in details in figure 1. you could not see the results is summarized in figure..'

3.the manuscript is disorganized and need to be re-written. ie. figure legends were inserted in the Results part. Please be careful. Legends for fig 4 and 5 were separated. Hardly to read.

4. Figure 2 A. What is the percentage means? what is the figure 2B telling us?

5. Figure 4A, is confusing, looks mismatch what depicted in the manuscript. What is' p16?' in the figure? is it unknown HPV? confused

6. Are those aberrant genetic features correlating with any patient survival?

6. PLOS authors have the option to publish the peer review history of their article (what does this mean?). If published, this will include your full peer review and any attached files.

Reviewer #1: No

Reviewer #2: No

---

## [Author Response · Author response to Decision Letter 0]

21 Jul 2020

Response to Editor and Reviewer:

Dear Editor,

We thank you very much for the comments and suggestions. The comments and suggestions are valuable and very helpful for revising and improving our manuscript. We have made revisions according to the referees’ comments and suggestions, as described in the authors’ response.

Additional Editor Comments (if provided):

Thank you for submitting this work to PLOS ONE. Please consider revising the manuscript as suggested by the reviewers.

Manuscript formatted according to PLOS ONE requirements.

2. Please provide additional details regarding participant consent. In the ethics statement in the Methods and online submission information, please ensure that you have specified what type of consent you obtained from patients for tissue collection (for instance, written or verbal, and if verbal, how it was documented and witnessed).

In addition, please provide the source of the tissue samples used in this work (e.g. hospital, institution or medical center name).

“All tissue samples were collected from subjects who gave informed consent using the Winship-Emory University IRB-approved consent form for tissue collection. Clinical data collected included gender, smoking history, radiation treatment status, chemotherapy treatment status and stage (AJCC 7)”.

3. Please provide the accession number of specific URL weblink of the specific RNA-Seq dataset downloaded from the TCGA for this study.

The following text added in Methods section

“TCGA RNA-seq data including 49 HPV-negative and 18 HPV-positive tissue samples in the form of raw gene count data was downloaded using TCGA2STAT package for R (disease=”HNSC” and data.type=”RNASeq2”) and used to find overlap between TCGA gene expression and our HNSCC data. Both expression data and clinical data were available for 67 HNSCC tumors (from different locations: oral tongue, base of tongue, and tonsil). We used the dataset abbreviations as defined by the TCGA consortium, a public resource that catalogues clinical data and molecular characterizations of many cancer types (as defined in https://tcga-data.nci.nih.gov/docs/publications/tcga/)”

Both RNA and DNA-Seq raw datasets were uploaded at NCBI SRA database (BioProject: PRJNA635454).

https://dataview.ncbi.nlm.nih.gov/object/PRJNA635454?reviewer=m305l851tkjcfjg7voodtmd3ma

Included the following text in the main manuscript 

“Data Availability: The sequencing data has been deposited at the Sequence Read Archive (SRA) under the BioProject ID PRJNA635454. All other relevant data are within the paper and its Supporting Information files.”

As you suggested by the editor we have removed the phrase “data not shown”.

6. Please upload a new copy of Figure 3 and 6 as the detail is not clear. Please follow the link for more information: https://blogs.plos.org/plos/2019/06/looking-good-tips-for-creating-your-plos-figures-graphics/

All 6 main and two supplementary figures were uploaded. 

Authors’ response

Reviewers' comments:

Reviewer's Responses to Questions

Comments to the Author

Reviewer #1: This work presents the signatures of somatic mutation and gene expression from p16ink4a positive HNSCC. Thank you for your valuable research on genetic variants and gene expression signatures of significant HPV infection in head and neck cancer. However, while reviewing this manuscript, it was difficult to check all the details because all the resolutions of the figures were low. Attaching high resolution figures will be helpful in the next review.

We have recreated all 6 main and two supplementary figure with high-resolution.

In this study, p16 of head and neck cancer FFPE was classified by IHC staining. In case of ‘p16 unknown’, didn't you perform IHC? Or did you stain IHC and the staining result wasn't clear? I recommend you need to define p16? group. I am still not sure why you compared p16 separately in this study. Therefore, the reason for comparing the p16 unknown group with the p16 positive group in this study should be clearly described in the result section.

We appreciate the author’s comments; we have attempted to check the p16 status on all samples; for some of the oral cavity samples residual tissue was exhausted; the lack of p16 status in these tumors even though a limitation does not alter the overall findings in this report given that these were oral cavity tumors. The fact that our results did not reveal notable differences between the p16 unknown and p16 negative is confirming of this as the clinical significance of a p16 positive status in oral cavity cancers is unclear. 

In addition, in the various experiments conducted in this study, some were targeted only to OPSCC, and some were confirmed in HNSCC, including cancers of a few other origins. Personally, I think it might have been clearer for this study to focus only on OPSCC. Or, in some experiments, please add the reason why only the OPSCC group was tested separately.

OPSCC and non-OPSCC are two subsites with in HNSCC.

Reviewer #2: comments:

1. figures are fuzzy, hardly to read

We have recreated all 6 main and two supplementary figure with high-resolution.

2. Please explain what you found in details in figure 1. you could not see the results is summarized in figure..'

Figure 1 summarizes the somatic mutation analysis with out using matching normal samples. Our results are consistent with previously analyses with matching normal samples (Genome Res 2019 (Ref 4); Nature 2015 (Ref 14)). 

3.the manuscript is disorganized and need to be re-written. ie. figure legends were inserted in the Results part. Please be careful. Legends for fig 4 and 5 were separated. Hardly to read.

We have reorganized according to PLOS One guidelines. 

Figure captions are inserted immediately after the first paragraph in which the figure is cited.

Tables are inserted immediately after the first paragraph in which they are cited.

4. Figure 2 A. What is the percentage means? what is the figure 2B telling us?

Percent sample size of total substitutions in a group (eg. p16-negative) from 151 target genes. Previous analysis of HPV-negative HNSCC data identified T>C substitutions were correlated with tobacco exposure (Genome Res 2019; Ref 4). HPV-negative samples were over represented compared to HPV-positive samples in both Genome Res 2019 (Ref 4) and Nature 2015 (Ref 14) papers. In our case, we have more HPV-positive samples compared to HPV-negative. 

5. Figure 4A, is confusing, looks mismatch what depicted in the manuscript. What is' p16?' in the figure? is it unknown HPV? confused

Figure 4A describes the differential expression analysis between p16-negative (Control) and p16-positive (Case) and, p16-unknown (Control) and p16-positive (Case) groups. Heatmap of commonly regulated/expressed genes suggested p16-unknown (Control) and p16-negative (Control) are similar in nature (added a new figure as supplementary file, Figure S2). 

As noted above, the p16 unknown cases were oral cavity cancers where the tissue was exhausted; we included these cases in our analysis as we think the behavior and profile of these tumors was likely the same as our p16 negative cohort. 

6. Are those aberrant genetic features correlating with any patient survival?

We have included patient survival analysis. From the curves, it is evident that the patients, who have negative (or unknown) status for the p16, have more death rate as compared to the patients, who are positive for p16. Moreover, a clear association of Kaplan-Meier survival curve of p16-based molecular subtype p16-negative with p16-unknown group supports RNA-seq based gene expression profiles (Figure. S2).

Nabil F Saba MD

Georgia Z Chen PhD

(Corresponding authors)

---

## [Decision Letter · Decision Letter 1]

19 Aug 2020

Signatures of somatic mutations and gene expression from p16INK4A positive head and neck squamous cell carcinomas (HNSCC)

PONE-D-20-10368R1

Dear Dr. Saba,

We’re pleased to inform you that your manuscript has been judged scientifically suitable for publication and will be formally accepted for publication once it meets all outstanding technical requirements.

Kind regards,

Hyunseok Kang, MD, MPH

Academic Editor

PLOS ONE

Additional Editor Comments (optional):

Reviewers' comments:

Reviewer's Responses to Questions

**Comments to the Author**

1. If the authors have adequately addressed your comments raised in a previous round of review and you feel that this manuscript is now acceptable for publication, you may indicate that here to bypass the “Comments to the Author” section, enter your conflict of interest statement in the “Confidential to Editor” section, and submit your "Accept" recommendation.

Reviewer #1: All comments have been addressed

Reviewer #2: All comments have been addressed

2. Is the manuscript technically sound, and do the data support the conclusions?

Reviewer #1: Yes

Reviewer #2: Partly

3. Has the statistical analysis been performed appropriately and rigorously? 

Reviewer #1: I Don't Know

Reviewer #2: I Don't Know

4. Have the authors made all data underlying the findings in their manuscript fully available?

Reviewer #1: Yes

Reviewer #2: Yes

5. Is the manuscript presented in an intelligible fashion and written in standard English?

Reviewer #1: Yes

Reviewer #2: Yes

6. Review Comments to the Author

Reviewer #1: All comments have been addressed. On discussion section you mentioned that 'Using RNA-seq data in parallel with targeted sequencing of a panel of 151 genes, we demonstrate that gene expression data from FFPE samples can identify gene signatures characteristic of p16 + versus p16- OPSCC which is a subsite of HNSCC.' but you used some of p16 non OPSCC samples (GHN-25). This paper analyzes data not only from OSCC but also from all HNSCC, and since data is mixed(p16+ all OSCC, p16- some OSCC and non-OSCC, and p16? all non-OSCC), accuracy of expression is considered to be important. Overall, if you care about this part, I think the rest of all is a well designed study.

Reviewer #2: The authors basically answered the question, although limitation for this research exist, it is qualified to be published.

7. PLOS authors have the option to publish the peer review history of their article (what does this mean?). If published, this will include your full peer review and any attached files.

Reviewer #1: No

Reviewer #2: No

---

## [Editor Report · Acceptance letter]

14 Sep 2020

PONE-D-20-10368R1 

Signatures of somatic mutations and gene expression from p16^INK4A^ positive head and neck squamous cell carcinomas (HNSCC) 

Dear Dr. Saba:

I'm pleased to inform you that your manuscript has been deemed suitable for publication in PLOS ONE. Congratulations! Your manuscript is now with our production department. 

Kind regards, 

on behalf of

Dr. Hyunseok Kang 

Academic Editor

PLOS ONE